# Epigenetic signature of human immune aging in the GESTALT study

Roshni Roy[1], Pei-Lun Kuo[2], Julián Candia[2], Dimitra Sarantopoulou[1], Ceereena Ubaida-Mohien[2], Dena Hernandez[3], Mary Kaileh[1], Sampath Arepalli[3], Amit Singh[1], Arsun Bektas[2], Jaekwan Kim[1], Ann Z Moore[2], Toshiko Tanaka[2], Julia McKelvey[4], Linda Zukley[4], Cuong Nguyen[5], Tonya Wallace[5], Christopher Dunn[5], William Wood[6], Yulan Piao[6], Christopher Coletta[6], Supriyo De[6], Jyoti Sen[7], Nan-ping Weng[1], Ranjan Sen[1], Luigi Ferrucci[2]*

[1]Laboratory of Molecular Biology and Immunology, National Institute on Aging, Baltimore, United States; [2]Translational Gerontology Branch, National Institute on Aging, Baltimore, United States; [3]Laboratory of Neurogenetics, National Institute on Aging, Bethesda, United States; [4]Clinical Research Core, National Institute on Aging, Baltimore, United States; [5]Flow Cytometry Unit, National Institute on Aging, Baltimore, United States; [6]Laboratory of Genetics and Genomics, National Institute on Aging, Baltimore, United States; [7]Laboratory of Clinical Investigation, National Institute on Aging, Baltimore, United States

*For correspondence:
ferruccilu@grc.nia.nih.gov

Competing interest: The authors declare that no competing interests exist.

**Abstract** Age-associated DNA methylation in blood cells convey information on health status. However, the mechanisms that drive these changes in circulating cells and their relationships to gene regulation are unknown. We identified age-associated DNA methylation sites in six purified blood-borne immune cell types (naive B, naive CD4[+] and CD8[+] T cells, granulocytes, monocytes, and NK cells) collected from healthy individuals interspersed over a wide age range. Of the thousands of age-associated sites, only 350 sites were differentially methylated in the same direction in all cell types and validated in an independent longitudinal cohort. Genes close to age-associated hypomethylated sites were enriched for collagen biosynthesis and complement cascade pathways, while genes close to hypermethylated sites mapped to neuronal pathways. In silico analyses showed that in most cell types, the age-associated hypo- and hypermethylated sites were enriched for ARNT (HIF1β) and REST transcription factor (TF) motifs, respectively, which are both master regulators of hypoxia response. To conclude, despite spatial heterogeneity, there is a commonality in the putative regulatory role with respect to TF motifs and histone modifications at and around these sites. These features suggest that DNA methylation changes in healthy aging may be adaptive responses to fluctuations of oxygen availability.

## Editor's evaluation

This fundamental work advances our understanding of chromatin changes that may be associated with aging across six distinct immune cell types. It highlights a non-uniform process of expression of aging signatures while a core signature is preserved across different cell types. The research employs solid validated and robust analysis methodologies. The findings would be of interest to researchers studying DNA methylation clock and aging biology.

## Introduction

Human aging is associated with site-specific changes of DNA methylation. Summary measures of DNA methylation called 'epigenetic clocks' are extensively used in aging research to estimate biological age (*Horvath and Raj, 2018*; *Hannum et al., 2013*; *Bocklandt et al., 2011*). Epigenetic clocks closely approximate chronological age and beyond age, predict adverse health conditions, including frailty (*Gale et al., 2018*), Alzheimer's disease (*McCartney et al., 2018*), and mortality (*Marioni et al., 2015*; *Chen et al., 2016*).

Research suggest that changes in DNA methylation with aging are regulated by specific mechanisms rather than by a stochastic drift (*Teschendorff et al., 2013*). For example, a loss-of-function mutation in the H3K36 histone methyltransferase has been associated with epigenetic aging in mice (*Martin-Herranz et al., 2019*). In humans, polymorphisms in the telomerase gene (TERT) (*Lu et al., 2018*) and age-dependent gain of methylation in the Polycomb repressive complex 2 have been related to accelerated aging (*Teschendorff et al., 2010*). However, so far, no sound hypothesis exists that explains the association of DNA methylation with aging and pathology.

A main obstacle in understanding mechanisms driving age-associated changes of DNA methylation is that most human studies were performed in mixed blood cell types. The few studies that investigated select immune circulating cells failed to propose a unifying biological hypothesis explaining predictable changes of DNA methylation with aging (*Dozmorov et al., 2017*; *Reynolds et al., 2014*; *Tserel et al., 2015*; *Bell et al., 2012*; *Kananen et al., 2016*; *Acevedo et al., 2015*; *Marttila et al., 2015*).

We analyzed age-associated methylation in six purified blood-borne cell types sorted from peripheral blood mononuclear cells (PBMCs) from 55 donors of ages ranging from 22 to 83 years. To minimize the confounding of age-associated pre-clinical and clinical diseases, participants were ascertained to be healthy by trained health professionals according to strict clinical criteria. We looked for CpGs differentially methylated with aging in the same direction in multiple cell types. Next, in each cell type, we conducted enrichment analyses of genes close to age-associated CpGs. Finally, we looked for chromatin accessibility markers and transcription factor (TF)-binding sites close to the same age-associated CpGs. Our findings suggest that changes in methylation with aging are related to fluctuation of energetic metabolism during the life course.

## Results

### Age-associated methylation in individual cell types

A principal component analysis (PCA) was performed on normalized DNA methylation data for all cell types from all the 55 donors (*Figure 1A* and *Supplementary file 1*). The PCA showed that clustering by cell types was stronger than by age (*Figure 1—figure supplement 1A*). The genes associated with the top 500 probes corresponding to PC1, PC2, and PC3 were enriched pathways linked to innate and adaptive lineage development (*Supplementary file 2*).

Age-associated CpGs were identified through sex-adjusted beta-regression models (FDR corrected p-value <0.05). The number of hypo- or hypermethylated sites varied considerably between cell types (*Figure 1B*) with highest numbers in CD4$^+$ T cells (*Figure 1—figure supplement 1B* and *Supplementary files 3 and 4*). Using a different approach of comparing between young (≤35 years, 25th percentile) and old (≥70 years, 75th percentile) individuals, we observed >90% overlap with beta-regression-derived hypomethylated sites and 70–95% overlap with hypermethylated sites in all cell types except CD8$^+$ T cells (9–14% overlap) (*Figure 1—figure supplement 1C*). Having fewer old donors with CD8$^+$ T cells may have contributed to differences (*Supplementary file 1*).

Like other studies, we found that a significant proportion of age-hypomethylated CpGs were in the intergenic and open sea (>4 kb from CpG island) regions while age-hypermethylated CpGs were in promoters and CpG islands (Chi sq test p < 0.001) (*Figure 1—figure supplement 1D, E*). Additionally, age-associated differentially methylated sites in PBMC poorly recapitulate age-dependent changes that take place in specific primary immune cells (*Figure 1—figure supplement 1E, F*). These findings point to a wide heterogeneity of age-differential CpG methylation across immune blood cells and suggest that studies in PBMC poorly represents the changes that take place in specific cell types with aging.

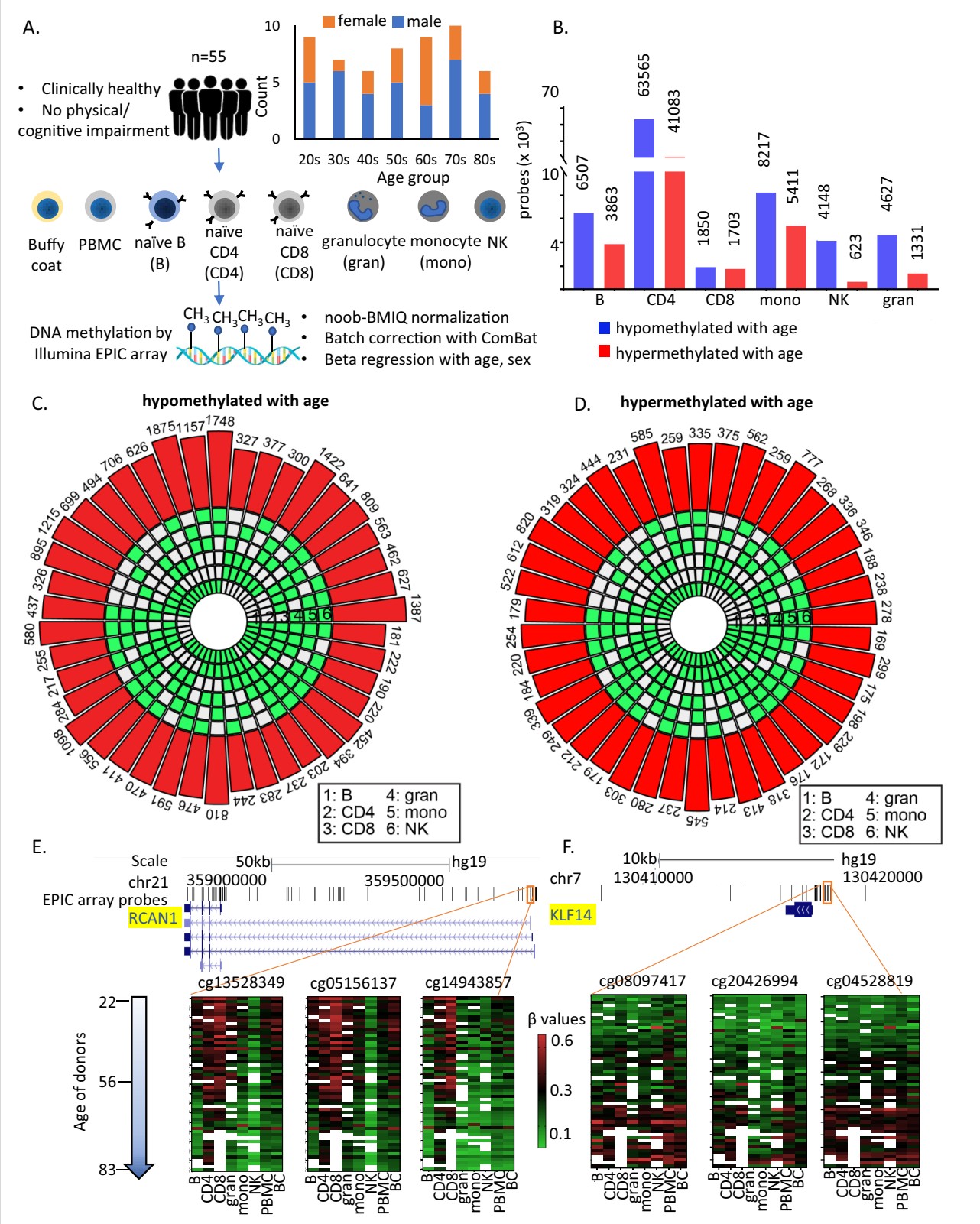

**Figure 1.** Study design and identification of age-associated methylation probes. (**A**) Study design. (**B**) Age-associated CpG methylation (False Discovery Rate or FDR p < 0.05) in six cell types. (**C, D**) SuperExactTest circular plots to show the number of age-associated hypo- and hypermethylated probes shared among different combinations of cell types (indicated by green boxes), respectively. The outermost bars show the number of probes shared among each cell-type combination (regardless of other cell types). For example, probes hypomethylated with age in B + CD4 + CD8 + gran + mono

*Figure 1 continued on next page*

*Figure 1 continued*

(n = 222) includes probes also hypomethylated in NK cells (n = 181) and probes not hypomethylated with age in NK cells (n = 41). Based on the exact probability distributions of multi-set intersections, all the overlaps shown are highly statistically significant (p < 10⁻¹⁰⁰). (**E**) Graphical representation of age-associated hypomethylation in promoter region of RCAN1 in all six cell types. (**F**) Graphical representation of age-associated hypermethylation in promoter region of KLF14. The methylation status in peripheral blood mononuclear cell (PBMC) and buffy coat are also shown. Missing methylation data are represented in white.

The online version of this article includes the following figure supplement(s) for figure 1:

**Figure supplement 1.** Characteristics of entire dataset and age-associated methylation data in six primary immune cells.

## Shared age-associated methylation across cell types

Only 181 age-associated hypomethylated sites and 169 hypermethylated sites were shared between all 6 cell types. These numbers increased to 776 (age-hypomethylated) and 404 (age-hypermethylated) sites in 5 or more cell types (*Figure 1C, D*). Thus, most age-related methylation changes are cell specific. Of note, only 10 of the sites overlap with the 359 CpGs in Horvath's pan-tissue epigenetic clock (*Horvath, 2013*). Several reasons can be attributed to this poor overlap including (1) use of methylation array with about 21,369 CpGs for development of the clock in contrast to the analyses in this study based on ~850,000 CpG sites; (2) use of data from peripheral or whole blood for development these clocks in contrast to data from flow-sorted circulating immune cells in this study. While the number of shared age-hypo- or hypermethylated CpGs across cells was relatively small, it was significantly much higher than that expected based on chance alone, suggesting that common underlying epigenetic mechanisms exist across the considered cell types (*Figure 1C, D*). For example, CpG sites adjacent to *RCAN1* (calcineurin 1) and *KLF14* (Krüppel-Like Factor 14) show similar age-associated patterns in all cell types (*Figure 1E, F*).

Next, we wanted to investigate whether the top age-associated genes are the ones which are shared across cell types. For this we arranged the age-associated probes with decreasing order of adjusted p-value and looked at the annotated genes to identify the top 15 genes in each cell type (*Figure 2A, B*, *Figure 2—figure supplement 1A–E*, and *Supplementary file 5*). THSD4 and CCDC102B were the most significant age-associated hypomethylated genes shared by five or more cell types, while ELOVL2, KLF14, LHFP14, and GPR158 were among the most significant age-hypermethylated genes in five or more cell types. This count increased to 5 and 13 genes, respectively, when the list was expanded to 50 top genes (*Supplementary file 5*). It is noteworthy that only 13–15% of these 'top' age-associated probes overlapped with the list of age-associated probes shared across cell types (181 hypomethylated and 169 hypermethylated probes). These findings suggest that most CpGs with age-associated methylation consistent across cell types undergo moderate (although significant) methylation changes with aging.

## Longitudinal validation of age-associated CpG sites

We hypothesized that the age-associated CpGs identified across the six immune cells in this cross-sectional study would also show longitudinal changes of the size and direction predicted. We used DNA methylation data (Illumina 450K microarray on DNA from buffy coats) assessed at baseline and 9- and 13-year follow-up in 699 participants of the InCHIANTI study (*Ferrucci et al., 2000*). Of the 181 hypomethylated and 169 hypermethylated CpGs with age in all cell types in GESTALT, 72 and 135, respectively, were represented in the 450K microarray (*Moore et al., 2016*). The beta-coefficients for age of the 207 CpG probes (72 + 135) estimated from the GESTALT study and their corresponding values estimated longitudinally from the InCHIANTI study were highly and significantly correlated (hypomethylated with age CpGs: r = 0.49, p = 1.2e−09 and hypermethylated with age CpGs: r = 0.5, p = 6.9e−06 for average beta coefficients across six cell types, *Figure 2C, D* and *Figure 2—figure supplement 2*). Thus, CpGs identified as differentially methylated with aging across cell types in GESTALT also change longitudinally with aging.

## Age-associated probes with opposite trends in different immune cells

Several CpGs showed significant but opposing age trends in different cell types, especially in B, CD4⁺ T cells, and monocytes (*Figure 2—figure supplement 1F, G*). For example, cg27123256 in the gene body of BCL11B was age hypomethylated in non-T cells and significantly age hypermethylated

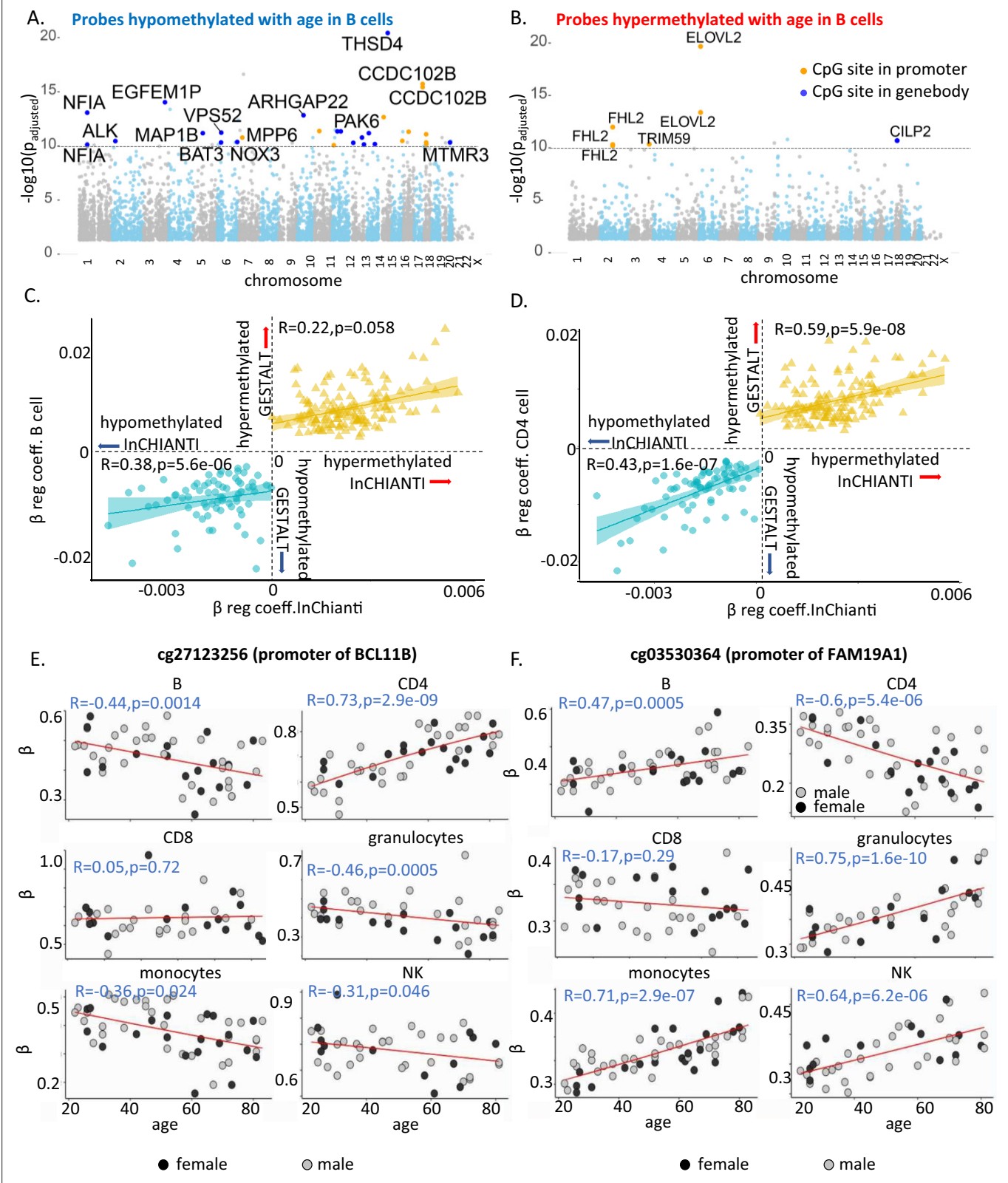

**Figure 2.** Characteristics of age-associated probes. (**A, B**) Manhattan plot of age-associated hypo- and hypermethylated CpG sites in B cells, respectively. Most significant genic probes (−log $p_{adj}$ >10) are labeled. (**C**) Correlation between beta-regression coefficients of age-differentially methylated CPGs in GESTALT and longitudinal InCHIANTI study. X-axis – InCHIANTI, Y-axis – B cell (**C**) and CD4$^+$ T cell coefficients (**D**). Blue dots – age-hypomethylated CpGs, yellow triangles – age-hypermethylated CpGs. (**E, F**) Scatter plot of age-associated CpGs showing opposite trends in different

*Figure 2 continued on next page*

*Figure 2 continued*

immune cells. (**E**) cg27123256 (in BCL11B promoter) is hypomethylated with older age in B, monocytes, and NK while is hypermethylated with older age in CD4[+] T cells. (**F**) cg03530364 (in FAM19A1 promoter) is hypermethylated with older age in B, granulocytes, monocytes, and NK cells while it is hypomethylated with older age in CD4[+] T cells.

The online version of this article includes the following figure supplement(s) for figure 2:

**Figure supplement 1.** Most significant age-associated CpGs in non-B immune cells along with CpGs showing opposite age-associated trends.

**Figure supplement 2.** Comparison of individual immune cells with InCHIANTI longitudinal study.

in naive CD4[+] T cells (*Figure 2E*). Our observations implicate BCL11B in aging-related changes in naive CD4[+] T cell function, distinct from its proposed role in effector cells (*Tserel et al., 2015*; *Gray et al., 2014*; *Yui and Rothenberg, 2014*). Conversely, cg03530364 in the body of FAM19A1 gene was hypermethylated in non-T cells but age-hypomethylated in CD4[+] T cells (*Figure 2F*). Of note, none of these CpGs were differentially age-methylated in PBMC. Thus, opposite age-methylation trends in specific cell types may cancel each other and obscure their relevance for aging when mixed cell-type samples are assessed.

## Pathway analysis of age-associated genes

Gene set enrichment analyses were performed on genes associated with at least one CpG significantly age-hypo- or hypermethylated in five or more cell types. We identified 30 pathways (*q*-value <0.05) (*Figure 3* and *Supplementary file 6*). Probes commonly age-hypomethylated in five or more cell types (*n* = 776) pointed to genes enriched in collagen biosynthesis, complement cascade, and GTPase pathways (left-most column in bottom panel of *Figure 3*) that highlighted inflammatory and metabolic pathway in aging. Genes associated with shared age-hypermethylated probes (*n* = 404) were enriched for neural pathways previously implicated to brain aging along with G-protein-coupled receptors pathways (*de Oliveira et al., 2019*; *Ewing et al., 2019*) (left-most column in top panel of *Figure 3*). A recent study by Karagiannis et al. also identified neuronal genes in their PBMC aging data emphasizing a possible interlink between immune-aging and neuronal pathways (*Karagiannis et al., 2023*). Other key pathways are highlighted, with associated genes displayed in boxes on the right-hand side.

## Functional annotation of age-associated probes

To further interrogate the relationships between DNA methylation and other epigenetic states, we mapped the methylation age-associated sites to cell-specific chromHMM-derived chromatin profiles (*Ernst and Kellis, 2012*). As controls, we annotated all sites in the EPIC array to the 18-state chromHMM model of respective primary cell type. Granulocytes were excluded from this analysis because reference data were not available.

Age-associated hypomethylated CpGs were significantly enriched for weak/active enhancers (yellow bar, *Figure 4A*) whereas, confirming previous reports, age-hypermethylated CpGs, were enriched in bivalent/polycomb regions compared to control set (brown and dark gray bars, respectively, in *Figure 4A*). Results for cell-type-specific analyses are shown in *Figure 4B*.

We further mapped the profile of four epigenetic markers from the ENCODE project in and around (±3 kb) age-associated methylation sites. For B and CD4[+] T cells, we observed a V-shaped peak-valley-peak pattern of DNase hypersensitivity at sites of age-associated hypomethylation, which is characteristic of promoter sites (*Figure 4C*; *Pundhir et al., 2016*). Both age-associated hypo- and hypermethylated sites showed evident H3K4me1 peaks, a marker commonly associated with active and primed enhancers (*Figure 4C*; *Bae and Lesch, 2020*). No specific trend was observed for H3K4me3 and H3K27ac (*Figure 4—figure supplement 1*). These patterns were highly consistent across cell types (*Figure 4—figure supplement 1*) and strongly suggest a functional connection between methylation and chromatin status. However, as the DHS and histone data in the ENCODE database were only available for either one of two donors (a 21-year-old male and 37-year-old female), we could not verify whether the patterns observed are stable with change in age.

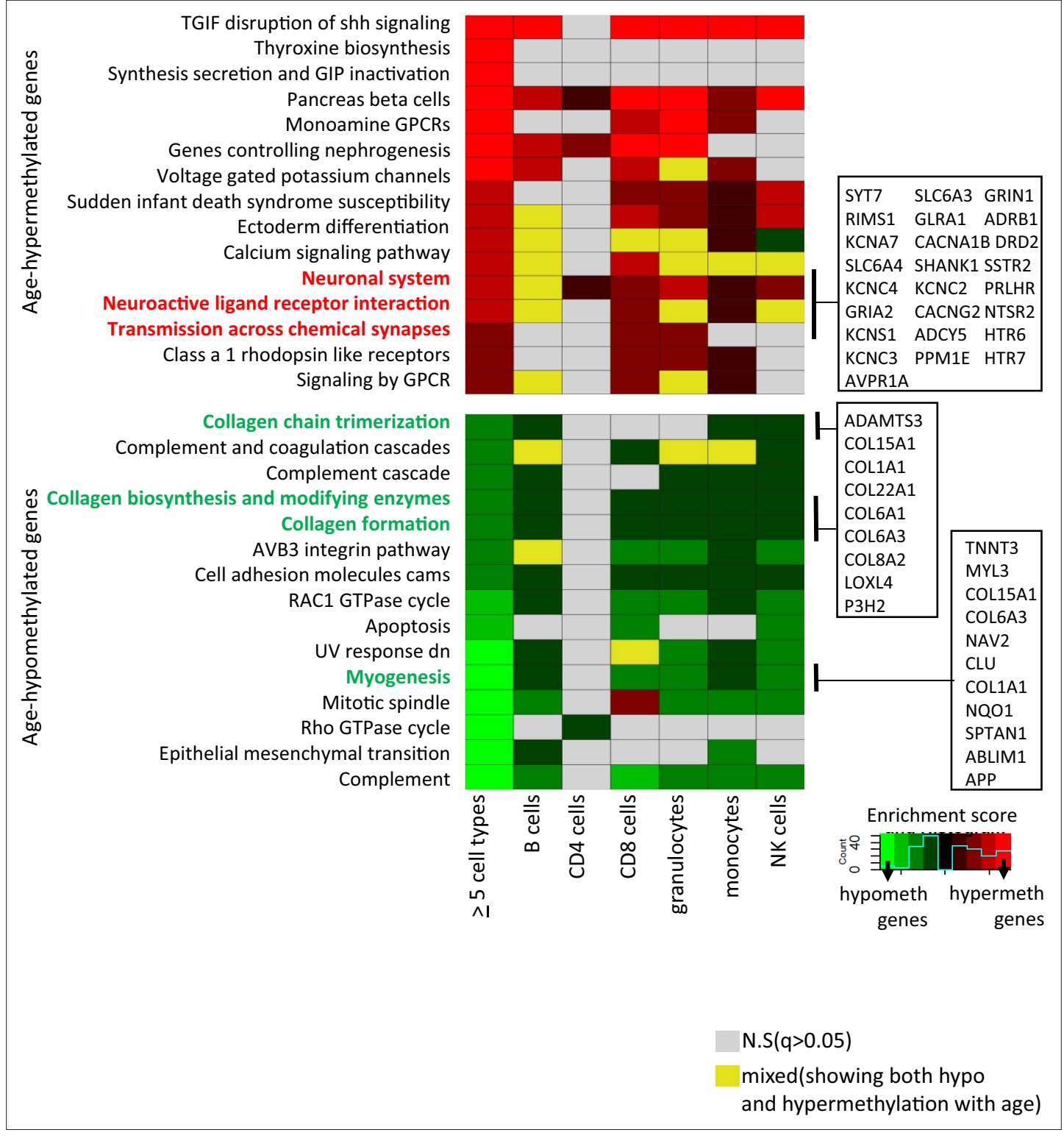

**Figure 3.** Pathway analysis of methylated probes. Enrichment analysis of genes annotated to age-associated hypo- and hypermethylated CpGs in ≥5 cell types (left-most column) and in individual cell types. Red/green shades indicate enrichment scores in hyper- (red) and hypo- (green) methylated genes. Yellow indicates ambiguous pathways associated with both hypo- and hypermethylated genes in individual cell types. Not significant pathways are shown in gray. Full results in ***Supplementary file 6***.

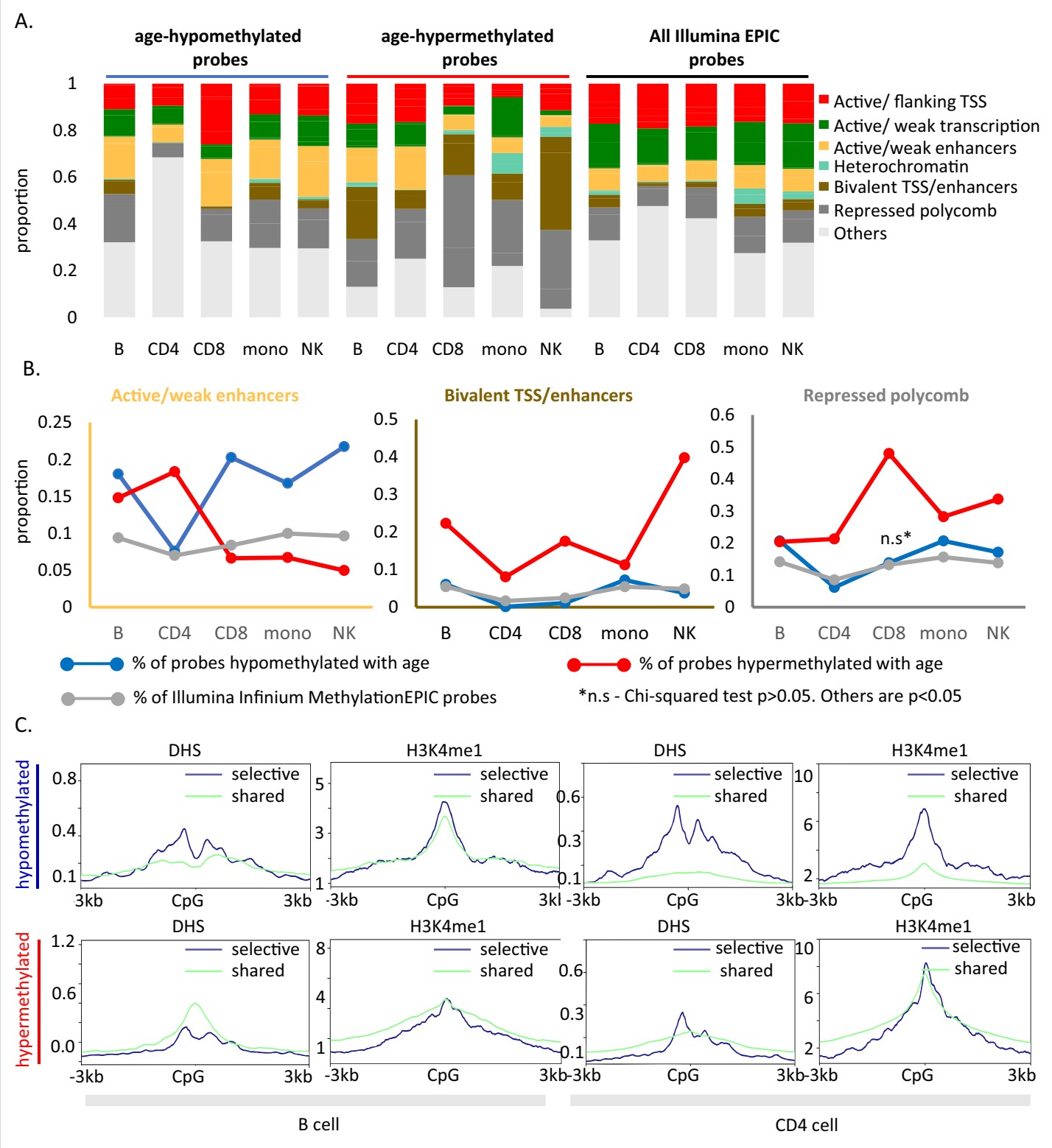

**Figure 4.** Functional annotation of age-associated probes along with their grouping based on sharedness. (**A**) ChromHMM annotation of age-associated CpGs. (**B**) Proportion of CpGs mapping to weak/active enhancers (left, orange box), bivalent enhancers/TSS (inset, brown box) and polycomb repressor regions (right, gray box) in age-associated hypo- (blue line), hypermethylated (red line) CpGs as compared to all MethylationEPIC CpGs (gray line). (**C**) DeepTools plots showing the distribution of accessible chromatin (DNase hypersensitive sites) and H3K4me1 histone mark in and around ±3 kb region of age-differentially methylated CpGs. The age-associated sites were divided into shared (blue) (common between five or more immune cells)

*Figure 4 continued on next page*

Figure 4 continued

and selective sites (green). The top row shows the pattern for age-associated hypomethylated CpGs while the bottom row is for the age-associated hypermethylated CpGs in B and CD4+ T cells.

The online version of this article includes the following figure supplement(s) for figure 4:

**Figure supplement 1.** Functional annotation of age-associated probes with respect to DHS and three other histone marks from ENCODE.

## Pattern of TF-binding motifs around age-associated CpGs

Specific TFs binding may induce loss of DNA methylation or bind DNA that is methylated (*Medvedeva et al., 2014*; *Moore et al., 2013*). Through our de novo HOMER analysis, we looked for TF-binding motifs in a 200-bp window around the age-associated methylated sites in each cell type. We observed that the binding motif for aryl hydrocarbon receptor nuclear translocator (ARNT, also named HIF1β) was associated with age-hypomethylated CpGs across most cell types (*Figure 5A*). The only exception was naive CD8+ T cells where the top enriched motif was B-cell lymphoma gene 6 (BCL6). BCL6 code for a zinc finger TF that plays a critical role in the generation of memory and effector cells in acute infection (*Kim et al., 2020*). Another motif associated with age-hypomethylated CpGs across most cell types was chromatin architectural protein CTCF and its closely related gene BORIS. Methylation changes at CTCF sites have been reported to reflect large-scale genome reorganization in immune cells in older individuals (*van Ruiten and Rowland, 2021*; *Bhat et al., 2021*).

Repressor Element 1-Silencing Transcription Factor (REST) was the TF motifs most frequently associated with age-hypermethylated CpGs in five of six cell types (*Figure 5B*). Age-hypermethylated sites in PBMCs have been previously shown to be enriched for REST, which is known to repress stress response genes and is lost in cognitive impairment and Alzheimer's disease pathology (*Yuan et al., 2015*; *Lu et al., 2014*). The top enriched TF motif associated with age-hypermethylated sites in monocytes was Arid5A ($p < 10^{-27}$) that binds to selective inflammation-related genes, such as IL6 and STAT3 and stabilize their expression (*Nyati et al., 2020*; *Wilsker et al., 2002*). We further repeated the analysis with a smaller 50 bp window size for TF motif search. Motifs for ARNT, CTCF, and REST remained the top hits in most cell types (*Supplementary file 7*). However, BCL6 and ARID5A were no longer the top motifs in the search indicating that motifs for these TFs appear to be farther from the age-associated CpG sites.

The recurring enrichment of ARNT and REST with age-associated CpGs observed across multiple cell types, despite relatively few shared genomic region locations, suggests a common mechanism of gene regulation. We found that only 17 and 44 age-associated hypo- and hypermethylated probes, respectively, shared ARNT or REST motifs across all cells (*Figure 5—figure supplement 1A, B*), suggesting these overlaps are not random and have a specific function (*Figure 5—figure supplement 1A, B*).

Remarkably, *ARNT* mRNA was significantly overexpressed in older age in three of the six cell types and *REST* mRNA showed a significant decrease of expression with age in most cell types (*Supplementary files 8 and 9*). These findings suggest that age-associated changes in expression levels of REST and ARNT can affect the epigenetic status of their target genes.

## Age-related differential methylation and oxygen sensing

ARNT, REST, and BCL6, three TFs most associated with differentially methylated regions, are implicated in hypoxia response (*Figure 5C*). ARNT is the beta subunit of Hypoxia Factor 1 (HIF-1), which is stabilized during hypoxia and shuttled to the nucleus where it binds to DNA hypoxia-response elements and triggers a complex response that include upregulation of angiogenesis and erythropoiesis and reprogramming of energetic metabolism from oxidative phosphorylation to anaerobic glycolysis (*Semenza, 2000*). Hypoxia also upregulates the transcription of REST which is the master regulator of the transcriptional repression arm of the response to hypoxia. Released REST is shuttled to the nucleus where it binds to DNA and regulates approximately 20% of the hypoxia-repressed genes, including genes involved in proliferation, translation, and cell cycle progression. We identified 35 genes that were hypomethylated with aging and had close by an ARNT motif in all six cell types (*Supplementary file 10*). Ten of these genes (right side of *Figure 5C*, genes under orange headings) have been linked to hypoxia response (*Craps et al., 2021*; *Stegmann, 1998*; *Chakraborty and Ain, 2017*; *Gusdon et al., 2012*; *Wang et al., 2009*; *Hsu et al., 2010*; *Pamenter et al., 2020*; *Pangou*

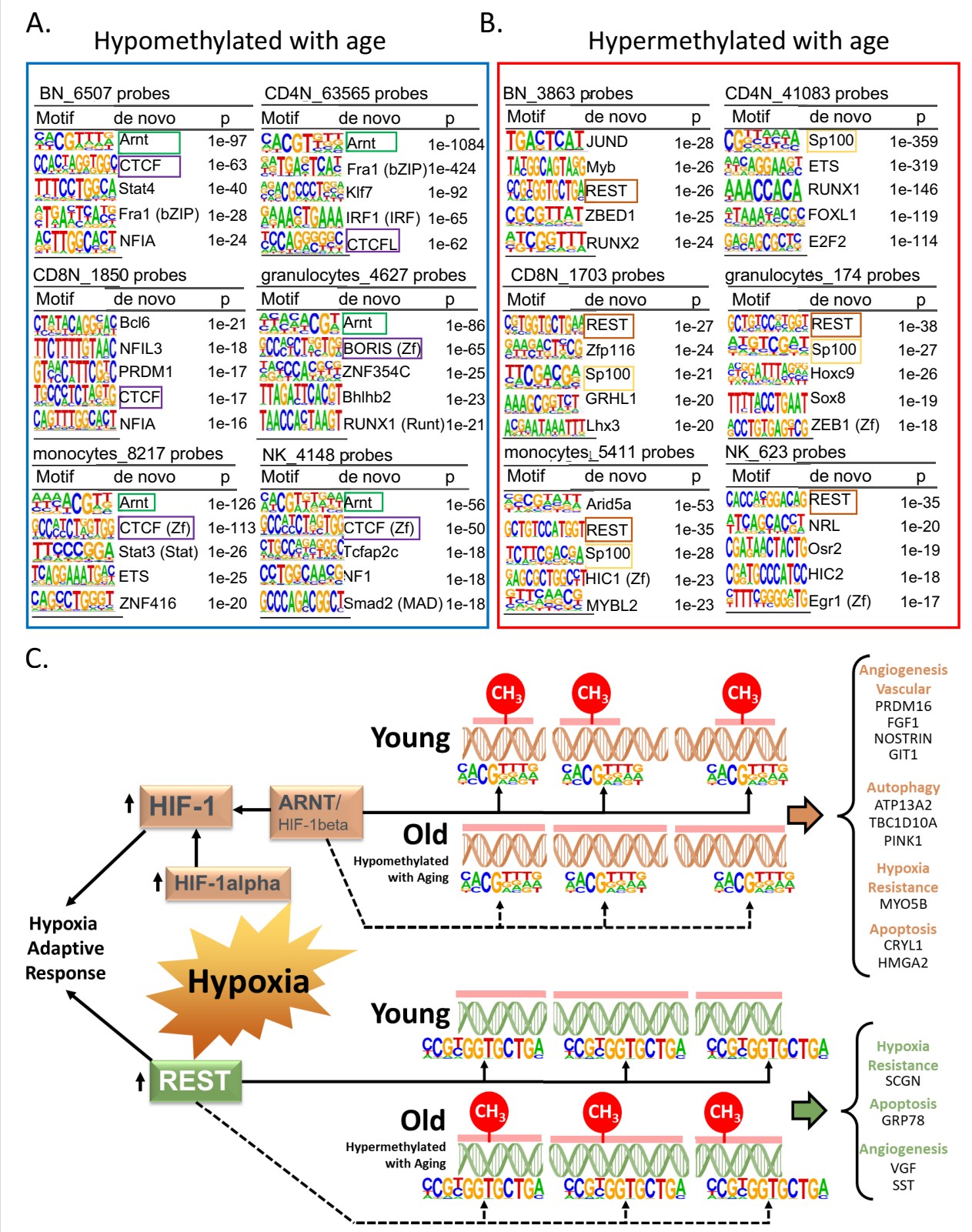

**Figure 5.** Association of transcription factor (TF)-binding motifs with age-differentially methylated CpGs. (**A**) Top 5 TF motifs at and around (±200 bp) of CpG sites that are hypomethylated with age. All the age-hypomethylated sites were considered for the analysis in each cell type. Recurring motifs like ARNT and CTCF/BORIS are highlighted. (**B**) Top 5 TF motifs at and around (±200 bp) CpG sites that are hypermethylated with age. All the age-hypermethylated sites were considered for the analysis in each cell type. Recurring motifs like REST and Sp100 are highlighted. (**C**) Hypoxia-centric

*Figure 5 continued on next page*

*Figure 5 continued*

model of age-associated sites with ARNT and REST motifs. CpG sites hypomethylated with aging across six different cell types are significantly more likely to host-binding motifs for ARNT, the core hub for the hypoxia response. On the contrary, CpG sites hypermethylated with aging are significantly more likely to host-binding motifs for REST, a hypoxia-response transcriptional repressor. On the right are selected age-associated genes that carry the motifs for ARNT or REST TFs.

The online version of this article includes the following figure supplement(s) for figure 5:

**Figure supplement 1.** Count of age-associated hypo- or hypermethylated probes with ARNT or REST motifs within 1 kb, respectively.

*et al., 2016*; *Lazarou et al., 2013*; *Cai et al., 2020*). Similarly, we found 26 genes with probes hypermethylated with age and with REST motif in the vicinity in all six cell types (*Supplementary file 10*). Four of these (right side of *Figure 5C*, genes under green heading) are known to be downregulated in hypoxia (*Tan et al., 2012*; *Liu et al., 2020*; *Dasgupta, 2004*; *Carmeliet and Jain, 2011*). These results strongly suggest a link between age-associated DNA methylation and oxygen sensing through putative regulation by TFs like ARNT and REST in the various immune cells.

## Association with inflammatory cytokines

Low-grade inflammation has been reported to be part of healthy aging. In order to investigate whether age-related pro-inflammatory state may explain the age-related changes in methylation observed in this study, we analyzed the SomaScan protein data of seven pro-inflammatory cytokines (IL6, IL1RN, IL1A, IL1B, TNF, TNFRSF1A, and TNFRSF1B) for the same cohort of donors from *Tanaka et al., 2018*. For each cell type, all CpGs reported as significantly hypo- or hypermethylated with age (in beta-regression analyses adjusted for sex) were reanalyzed by incorporating data on seven pro-inflammatory cytokines (see Materials and methods for details). Briefly, by comparing a model with a cytokine as explanatory variable (CpG ~ age + sex + cytokine) with another model without it (CpG ~ age + sex), we explored the robustness of age as an explanatory variable of methylation change, as well as possible mediating effects arising from pro-inflammatory cytokines. Detailed results are provided in OSF and summary statistics for each cell type, hypo-/hypermethylation association and pro-inflammatory cytokine are provided in *Supplementary file 11*. By comparing the results from the two abovementioned regression models, we observed in case of hypermethylated sites in CD4 cells, the number of CpGs dropped by 10% on adding TNFRSF1A to the model, a cytokine that appears significantly associated with 7507 of those ge-associated CpGs (Column I). In addition, TNFRSF1B appears significantly associated with 9058 of the age-hypermethylated CpG sites in CD4 cells. For other cell types like B naive and monocytes, TNF-alpha was associated with 65–124 age-associated CpG sites, respectively. Fewer associations are observed for the remaining analytes. These results suggest a possible link between TNF-alpha signaling pathway, aging, and DNA methylation change in circulating immune cells.

## Discussion

Novel and important conclusions arise from our observations. First, only few CpG sites are hypo- and hypermethylated with aging across all circulating cells while majority of the significant age-associated methylation changes are cell selective. Indeed, several CpGs show differential age methylation in opposite directions in different cell types and are unchanged in PBMC, suggesting that they may be missed when studying mixed cell samples. Noteworthy, age-related methylation differences in this cross-sectional study were strongly and significantly correlated with longitudinal age-associated methylation changes in an independent population.

Second, age-associated hypomethylated sites were significantly enriched for active enhancers whereas age-hypermethylated sites were enriched for bivalent/polycomb regions, confirming previous findings in whole blood (*Yuan et al., 2015*). Age-differential methylation coincided with specific chromatin status and histone markers patterns, suggesting that their position in proximity of promoter and active enhancer regions is connected with chromatic accessibility and potentially modulation of gene expression. Since the ENCODE data were only from two donors, it will be worthwhile to see how the histone or chromatin accessibility patterns change with age at and around these age-associated CpG sites.

Third, distinct TF-binding motifs co-localize with CpGs differentially methylated with aging despite wide variation in the distribution of such sites across cell types, suggesting a specific regulatory function. Noteworthy, the top age-associated TF identified, ARNT and REST act in coordination in hypoxia response (*Cavadas et al., 2016*). BCL6, another top TF-binding motif associated with age-differentially methylated CpG has also been shown to protects cardiomyocyte from damage during hypoxia (*Gu et al., 2019*). These findings support the hypothesis that systematic methylation changes with aging may be induced by fluctuations in oxygen availability and energy metabolism. Interestingly, the mRNA encoding *ARNT* significantly increases with age in all cell types except monocytes, while mRNA coding for *REST* declines with aging in four cell types and shows no significant change in naive CD8[+] T cells and NK cells. mRNAs coding for *CTCF* showed strong age association across numerous cell types (*Supplementary file 8*). The hypothesis that oxygen sensing regulates directly or indirectly DNA methylations is consistent with studies showing that in replicating fibroblasts, biological age estimated by DNA methylation slows down under hypoxia compared to normoxia (*Matsuyama et al., 2019*). Further, many genes close by to 'shared' age-differentially methylated CpG identified in our analyses play important roles in hypoxia response (*Figure 5C*).

The specific mechanisms connecting age-related changes in DNA methylation in genes which also contain binding motifs the master hypoxia-response mediators remain unknown. Shahrzad et al. reported an inverse correlation between the severity of hypoxia and the degree of DNA methylation (*Shahrzad et al., 2007*). There is evidence that hypoxia-induced hypermethylation may be due to reduced TETs activity (*Thienpont et al., 2016*). Our findings add to this literature by suggesting that a direct interaction between hypoxia-related TFs and DNA methylation at specific DNA sites occur with aging, perhaps as an adaptive response triggered by fluctuations in oxygen levels that occur in many age-related conditions. This hypothesis is consistent with oxygen availability been the most important environmental factor that requires physiological adaptation during pregnancy and development and extends this concept in a life course perspective.

A limitation of this study is that we have focused on circulating cells and, therefore, our findings may not apply to age methylation in other tissues. In addition, our findings were not replicated in an independent cross-sectional study population. Despite these limitations, this study has unique features: a cohort of exceptionally healthy donors and percent methylation was assessed in specific cell types obtained by cytapheresis and sorted by using state-of-the art methods.

## Conclusion

Age-associated DNA methylation profiles of the six purified primary immune cell populations in the blood show more cell specificity than sharedness. However, we observe common regulatory features with respect to TF-binding motifs and histone modifications. Based on the consistent association of these methylated sites with ARNT and REST, which are master hypoxia regulators, we hypothesize that oxygen sensing and hypoxia drive mechanisms for changes in methylation. This hypothesis should be further explored in animal models with manipulation of oxygen levels and serial measures of DNA methylation in circulating immune cells.

# Materials and methods
## Cohort details

Buffy coat, PBMCs, and granulocytes were collected from Genetic and Epigenetic Signatures of Translational Aging Laboratory Testing study (GESTALT) study participants (*N* = 55; 34 men and 21 women; age 22–83 years) who were free of diseases (except controlled hypertension or history of cancer silent for >10 years), not on medications (except one antihypertensive drug), had no physical or cognitive impairments, non-smokers, weighed >110 lbs, had body mass index <30 kg/m$^2$ (*Roy et al., 2021*; *Ubaida-Mohien et al., 2019*). GESTALT was approved by the institutional review board of the National Institutes of Health and participants explicitly consented to participate.

## Isolation of PBMC and immune cell populations

PBMCs were isolated from cytapheresis packs by density gradient centrifugation using Ficoll-Paque Plus. Total B, CD4[+], and CD8[+] T cells were enriched by negative selection using EasySep Negative Human kits specific for each cell type; monocytes were negatively enriched using 'EasySep Human

Monocyte Enrichment Kit w/o CD16 depletion'. Natural killer cells were negatively enriched by depleting PBMCs with antibodies against CD3, CD4, CD14, CD19, and Glycophorin-A in HBSS (Hanks' Balanced Salt Solution) buffer. Enriched cell populations were FACS (fluorescence-activated cell sorting) sorted by flow cytometry as per Human Immunophenotyping Consortium (HIPC) phenotyping panels (*Maecker et al., 2012*). Gating strategies and post-sort purity were analyzed by FlowJo software (LLC, Ashland, OR) (*Roy et al., 2021*). Granulocytes were positively selected from whole blood using EasySep Human Whole Blood CD66b Positive Selection Kit. Purified cells and PBMC were washed with phosphate-buffered saline, snap frozen and stored at −80°C. All sorted cells were >95% pure by flow cytometry (*Roy et al., 2021*).

## Assessment of DNA methylation

DNA was isolated from 1 to 2 million cells using DNAQuik DNA Extraction protocol and the Qiagen DNeasy Kit. 300 ng of DNA was treated with sodium bisulfite using Zymo EZ-96 DNA Methylation Kit. The methylation of ~850,000 CpG sites was determined using Illumina Human MethylationEPIC BeadChip, and data preanalyzed by GenomeStudio 2011.1.

## Data processing and functional annotation of CpG sites

Analyses were performed by the R minfi package (*Aryee et al., 2014*; *Fortin et al., 2017*). Probes with low detection p-values (cutoff 0.01) were filtered out (*Moran et al., 2016*). Data were normalized using noob and BMIQ (*Liu and Siegmund, 2016*), batch corrected by ComBat function (sva package), and $\beta$ values were used for differential methylation analyses. Following the MethylationEPIC probe annotation (IlluminaHumanMethylationEPICanno-.ilm10b2.hg19) to the UCSC RefSeq genes (hg19), we grouped the locations into three categories: (1) promoter group – TSS 1500 (from 201 to 1500 bp upstream of TSS), TSS 200 (≤200 bp upstream of TSS), 5′UTR, first exon; (2) genebody – exons (all exons except exon1), exon intron boundary, intron and 3′UTR; and (3) intergenic probes. The first gene in the annotation package was considered. Probes were divided into three groups – within CpG islands (CGI), within CpG shore (0–2 kb from CGI), CpG shelf (2–4 kb from CGI), and open sea (>4 kb from CGI).

## Definition of age-associated probes

Age- and sex-adjusted CpG-specific beta-regressions were performed on normalized $\beta$ values using the R *betareg* function. p-values were adjusted for multiple testing (Benjamini–Hochberg [BH] adjusted p < 0.05). Probes with FDR p < 0.05 for age and FDR p > 0.05 for sex were considered age-differentially methylated CpGs. Beta-regression estimate value was used to group the age-associated probes as hypo- (Estimate$_{age}$ <0) or hypermethylated (Estimate$_{age}$ >0). The overlap of probes across multiple combinations of the six cell types was assessed using R package SuperExactTest (v.1.1.0) (*Wang et al., 2015*).

## Gene set enrichment analysis

Based on the EPICarray annotation, genes were classified as differentially hypo- or hypermethylated with age. Genes with both age hypo- and hypermethylated CpGs were removed from the analysis. Enrichment analysis was performed by the tmodHGtest method in the tmod v.0.46.2 R package, comparing a foreground list of genes found in ≥5 cell types against reference gene set collections 'Hallmarks' and 'Canonical Pathways' (which includes Reactome, KEGG, WikiPathways, PID, and Biocarta gene sets) from the Molecular Signature Database MSigDB (v.7.4) (*Subramanian et al., 2005*).

For the gene enrichment analysis of the principal components, the top 500 CpG probes corresponding to the positive and negative directions along PC1, PC2, and PC3 were extracted and annotated to nearest gene as per the manufacturer's annotation file. The ambiguous genes with probes associated with both positive and negative PC directions were removed from the analysis. The remaining genes were run through the abovementioned enrichment analysis pipeline. A filter based on *q*-value <0.05 was imposed to find the most significant pathways.

## Visualization of histone peaks and DHS peaks

Primary cell DHS and chromatin ChIP-Seq bigwig files were downloaded from ENCODE (*Roy et al., 2021*). DeepTools was used to visualize DHS and histone peaks in +3 kb region surrounding

age-associated shared and non-shared methylated sites. For plotting purposes, the order of methylated probes was determined based on descending score of DHS peaks and followed for all histone marks (H3K4me1, H3K4me3, and H3K27ac).

## Annotation of age-associated methylated probes using chromHMM
The 18-state chromHMM models (based on 6 chromatin marks H3K4me3, H3K4me1, H3K36me3, H3K27me3, H3K9me3, and H3K27ac) for various immune cells (E032 – primary B cell, E038 – primary naive CD4[+] T cells, E047 – primary naive CD8[+] T cells, E029 – monocyte, E046 – NK cell) were downloaded from Roadmap epigenomics project. Bedops tool was used to map the age-associated methylated sites to the respective chromHMM profiles. All Infinium MethylationEPIC array probes were also partitioned using each of the immune cell chromHMM profiles as controls.

## Prediction of de novo TF-binding motifs by HOMER
All the age-associated methylation sites were considered for HOMER analysis. A region of ±200 bp around each age-associated methylated site was provided as input for analysis in HOMER using de novo setting (*Heinz et al., 2010*). As a background, we used the default background list that HOMER creates by matching the GC% in the input list. The output from the stringent de novo analysis was considered for downstream data interpretation.

## InCHIANTI longitudinal study cohort
InCHIANTI (Invecchiare in Chianti) is a population-based cohort of individuals ≥20 years old from the Chianti region of Tuscany, Italy (PMID: 11129752). The Italian National Institute of Research and Care on Aging Institutional Review Board approved the study protocol and all participants explicitly consented to participate. DNA methylation from 699 participants (1841 observations) was used for the analysis. CpG methylation of 485,577 CpGs was determined by the Illumina Infinium HumanMethylation450 BeadChip (Illumina Inc, San Diego, CA) and data processed by the R package 'sesame'. Mean rates of change were estimated from 2 to 3 longitudinal timepoints.

## RNA-Seq sample extraction, processing, and data analysis
Total RNA was extracted from $2 \times 10^6$ cells, depleted from ribosomal RNA and 50 ng was used for cDNA synthesis and library preparation. Libraries were sequenced for 138 cycles on Illumina HiSeq 2500. After adapter removal and end trimming of raw FASTQ files, transcript abundances were quantified with reference to hg19 transcriptome using kallisto 0.44 (with options --single -l 250 -s 25). Transcripts were aggregated to genes with tximport and filtered out if less than 10 TPM were detected in more than 33% of the samples. Linear regression models (~phase + age*sex) were used on TPM normalized expression values to study expression changes of selected TFs with age. Only the regression coefficient and p-value for the three TF genes – *ARNT*, *CTCF*, and *REST* were used in this study.

## Inflammatory cytokine analysis
Published SomaScan protein data from the same cohort of donors used in the present study were extracted to look for age-associated changes in seven cytokines (IL6, IL1RN, IL1A, IL1B, TNF, TNFRSF1A, and TNFRSF1B) (*Tanaka et al., 2018*). Briefly, plasma proteomics was measured using the 1.3k SomaScan assay (SomaLogic, Boulder, CO) followed by standard quality control and normalization procedures as described in previous publications (*Candia et al., 2017*; *Candia et al., 2022*). Normalized data for seven cytokines were extracted (detailed annotation provided in *Supplementary file 10*). To complement the age association analysis of CpGs adjusted by sex (CpG ~ age + sex), we performed additional beta-regression analyses separately including each target pro-inflammatory cytokine as an explanatory variable in the form: CpG ~ age + sex + cytokine. Details about the cytokines are provided in OSF.

## Acknowledgements

This work was supported entirely by the Intramural Research Program of the National Institute on Aging. We are grateful to the GESTALT participants and the GESTALT Study Team at Harbor Hospital and NIA.

# Additional information

## Funding

No external funding was received for this work.

## Author contributions

Roshni Roy, Luigi Ferrucci, Conceptualization, Formal analysis, Supervision, Funding acquisition, Visualization, Methodology, Writing – original draft, Writing – review and editing; Pei-Lun Kuo, Data curation, Formal analysis, Writing – review and editing; Julián Candia, Formal analysis, Writing – original draft; Dimitra Sarantopoulou, Formal analysis, Methodology, Writing – review and editing; Ceereena Ubaida-Mohien, Software, Visualization; Dena Hernandez, Data curation, Supervision, Methodology, Writing – review and editing; Mary Kaileh, Conceptualization, Resources, Methodology, Project administration, Writing – review and editing; Sampath Arepalli, Jaekwan Kim, Toshiko Tanaka, Tonya Wallace, Christopher Dunn, William Wood, Christopher Coletta, Data curation; Amit Singh, Visualization, Methodology, Writing – review and editing; Arsun Bektas, Data curation, Writing – review and editing; Ann Z Moore, Data curation, Formal analysis; Julia McKelvey, Linda Zukley, Conceptualization, Data curation; Cuong Nguyen, Yulan Piao, Data curation, Methodology; Supriyo De, Software, Formal analysis, Methodology; Jyoti Sen, Supervision, Methodology, Writing – review and editing; Nan-ping Weng, Conceptualization, Methodology, Writing – review and editing; Ranjan Sen, Conceptualization, Supervision, Writing – review and editing

## Author ORCIDs

Julián Candia ⓘ http://orcid.org/0000-0001-5793-8989
Ceereena Ubaida-Mohien ⓘ http://orcid.org/0000-0002-4301-4758
Mary Kaileh ⓘ http://orcid.org/0000-0003-2314-312X
Toshiko Tanaka ⓘ http://orcid.org/0000-0002-4161-3829
Christopher Dunn ⓘ http://orcid.org/0000-0001-7899-0110
Luigi Ferrucci ⓘ http://orcid.org/0000-0002-6273-1613

## Ethics

GESTALT study was approved by the institutional review board of the National Institutes of Health. Informed consent as well as the consent to publish the data collected was obtained from every participant in the study. Since the study of gene expression and epigenetic regulation are essential aims of GESTALT, all participants were required to consent to DNA/RNA testing and storage at all visits in order to participate in the study. The GESTALT IRB approval number is 15-AG-0063.

## Decision letter and Author response

Decision letter https://doi.org/10.7554/eLife.86136.sa1
Author response https://doi.org/10.7554/eLife.86136.sa2

# Additional files

## Supplementary files

• Supplementary file 1. Demographic and flow cytometry marker details of the cohort. Details of the age and sex distribution of the healthy donors from the GESTALT study for each of the primary immune cell-type population are described. The flow cytometry markers for cell selection are also mentioned.

• Supplementary file 2. Pathway enrichment analysis of genes annotated to top 500 probes corresponding to PC1, PC2, and PC3 components of principal component analysis (PCA). The top 500 CpG sites corresponding to PC1, PC2, and PC3 components were annotated to genes followed by gene enrichment analysis. Age-associated genes in each pathway are in column M.

- Supplementary file 3. Distribution of slope for probes significantly changing with age in the immune cells. The age-associated probes were identified from beta-regression (FDR p < 0.05).

- Supplementary file 4. List of age-associated probes each of the six primary immune cells. Beta-regression coefficient, FDR p-value, and genomic annotation of the age-associated probes were identified from beta-regression (FDR p < 0.05).

- Supplementary file 5. List of top age-associated genes in the six immune cell types. The list of top 15 and top 50 age-associated hypo- and hypermethylated genes derived from the most significant age-associated probes in each cell type.

- Supplementary file 6. Detailed output of gene set enrichment analysis. Gene set enrichment analysis was performed on genes based on annotation of age-associated hypo- and hypermethylation probes commonly changing in five or more cell types.

- Supplementary file 7. Top 5 transcription factor (TF) motifs within ±50 bp of age-associated methylated sites. HOMER de novo analysis was performed to identify the top 5 TF motifs within 50 bp of age-associated hypo- and hypermethylated sites in each of the six cell types.

- Supplementary file 8. Average read depths and Kallisto TPM normalized read counts of ARNT, CTCF, and REST for all the donors. RNA-Seq data were used to look into the gene expression change of three selected transcription factors (TFs; ARNT, CTCF, and REST) with age. These TF motifs are most commonly associated with the age-related methylated sites in all immune cells. The mapping rates along with the Kallisto TPM normalized values for the three TFs for each cell type in each of the donors have been shown.

- Supplementary file 9. Age-associated differences of transcripts for ARNT, REST, and CTCF. FDR p-values derived from the linear regression of expression levels of the three transcription factors (TFs) with age in each of the six cell types.

- Supplementary file 10. List of genes with age-associated methylated CpG sites showing ARNT or REST motif within 1 kb. The age-associated probes with ARNT or REST motifs within 1 kb region were annotated to genes and summarized into a table. For each gene, number of age-associated CpG sites with ARNT/REST motif and number of cell types in which this occurrence has been observed have been mentioned.

- Supplementary file 11. Output of beta-regression analysis with age and sex and the seven analytes. Summary of two beta-regression models has been tabularized. Column C shows the number of age-associated probes from the original model CpG ~ age + sex with FDR cutoff of adjusted $p_{age}$ < 0.05. Columns D–J show the number of age-associated probes from the model CpG ~ age + sex + analyte with FDR cutoff of adjusted $p_{age}$ < 0.05 and adjusted $p_{analyte}$ < 0.05. Finally, columns K–Q represent the number of age-associated probes from the model CpG ~ age + sex + analyte with FDR cutoff of adjusted $p_{analyte}$ < 0.05.

- Supplementary file 12. List of softwares.

- MDAR checklist

## Data availability

Researchers interested in using the data from the previously published InCHIANTI study are invited to submit a proposal for consideration, for full details please see https://www.nia.nih.gov/inchianti-study. Code and data processing scripts (including a de-identified version of the GESTALT dataset) are available on OSF. DNA methylation EPIC 850k data are available at GEO under accession number GSE184269.

The following dataset was generated:

| Author(s) | Year | Dataset title | Dataset URL | Database and Identifier |
| --- | --- | --- | --- | --- |
| Roy et al | 2023 | Epigenetic signature of human immune aging: the GESTALT study | https://osf.io/rxw6h/ | Open Science Framework, rxw6h |

The following previously published dataset was used:

| Author(s) | Year | Dataset title | Dataset URL | Database and Identifier |
|---|---|---|---|---|
| Kaileh M, Roy R, Ramamoorthy S, Boller S, Grosschedl R, De S | 2021 | Specification of human immune cell epigenetic identity by combinations of transcription factors (MethylationEPIC) | https://www.ncbi.nlm.nih.gov/geo/query/acc.cgi?acc=GSE184269 | NCBI Gene Expression Omnibus, GSE184269 |

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
