## [Editor Report]

This fundamental work advances our understanding of chromatin changes that may be associated with aging across six distinct immune cell types. It highlights a non-uniform process of expression of aging signatures while a core signature is preserved across different cell types. The research employs solid validated and robust analysis methodologies. The findings would be of interest to researchers studying DNA methylation clock and aging biology.

---

## [Decision Letter]

**Decision letter after peer review:**

Thank you for submitting your article "Epigenetic signature of human immune aging: the GESTALT study." for consideration by *eLife*. Your article has been reviewed by 3 peer reviewers, one of whom is a member of our Board of Reviewing Editors, and the evaluation has been overseen by Carlos Isales as the Senior Editor. The reviewers have opted to remain anonymous.

Essential revisions:

1) A number of points require clarification as indicated by the reviewers. Please address each of these points in revision.

2) Clarify the study is a reanalysis of previously published data to shed light on new patterns.

*Reviewer #1 (Recommendations for the authors):*

This is an interesting study that potentially identifies a single fundamental mechanism that regulates chromatin changes in aging from stochastic changes. The notion that the methylation changes are cell selective is an interesting concept.

Building on the comments regarding circulating and tissue-resident cells, can the authors provide data to support or refute whether programs are consistent in the different compartments? It would strengthen the work to provide discussion on these points.

The interpretation of the data relies on changes in genes related to hypoxia to support that the metabolic state of the cell determines outcomes. It would be useful to test this hypothesis which would be possible in vitro to determine if programs are indeed determined by metabolic states that in vivo differ in different microenvironments, or that cells are intrinsically differentially programmed.

Although the samples were obtained from participants who were deemed 'clinically' healthy, were any finer criteria used to distinguish and cross-correlate the immune cell phenotype with the methylation changes? The identification of an inflammatory program in the data associated with aging is interesting. While it might occur that low-level inflammation is inherently associated with aging, distinguishing the two programs would be important in teasing the data apart. Thus, a current limitation of the work is the lack of cellular phenotypic and functional mapping corresponding to the molecular analysis to bring clarity to whether overlapping programs occur and inflammation contributes to driving aging normally, or whether separate programs usually occur in parallel but could overlap. Dissecting this apart further would strengthen the work.

*Reviewer #2 (Recommendations for the authors):*

1. The authors claimed that clustering by cell type (PC2) was stronger than by age. For the PCA, it would be great if the authors could explain what PC1 indicates and present the information about all PCs.

2. It has been reported that sex-related differences contribute to immune cell aging (Huang et al., PNAS, 2021, PMID:34385315; Marquez et al., Nature Communications, 2020, PMID: 32029736). It is unclear how sex differences affect the dataset that the authors analyzed and how sex-adjusted β-regression was conducted.

3. The authors identified 350 age-associated differentially methylated sites among all six immune cell types and compared them with other published studies. Do these sites overlap with the DNA methylation clock sites identified in PBMCs since these sites change in the same direction in separate cell populations? Also, only 10 of these 350 sites overlap with Horvath's pan-tissue epigenetic clock. Is there any factor that can explain this difference? For example, sequencing depth, coverage, etc?

4. In the legend of Figure 2, the most significant genic probes are classified as -log10(padj)<10. Did the author mean -log10(padj)>10?

5. It needs to be clarified how the top 15 genes were chosen. How many age-associated CpGs contribute to these 15 genes? Also, if more genes are considered, will more genes be shared among cell types?

6. In Figure 3C and 3D, the differences in β coefficient in the InCHIANTI study are small (scale at 0.003 and 0.006). Are the differentially methylated sites called in GESTALT identified as differentially methylated in the original InCHIANTI study?

7. In Figure 4C and Supp Figure 4, it seems both hypo- and hyper-methylated sites with age show the V-shaped pattern.

8. It is unclear how the expression levels were measured. Expression levels from qPCR or RNA-seq need to be shown in addition to Supp Table 6. RNA-seq was mentioned in the method section, and if this was performed in this study, the quality metrics for the data need to be provided and the data need to be uploaded to GEO. The current GEO number (GSE184269) provided in this manuscript is linked to a published study and does not contain RNA-seq data.

9. There are several places where the authors mentioned "data not shown". It may be better if these data could be presented in supplementary figures or tables.

*Reviewer #3 (Recommendations for the authors):*

Specific suggestions for Authors:

1) As mentioned in the public review, certain claims should be toned down or placed in context.

2) The authors need to make it clear in the introduction that this current study is based on the reanalysis of a data set they have published in 2021 and describe key findings and limitations of the initial study.

3) It is not clear from the methods and figure legends that motif predictions were done on DMRs that could be found in at least 5 cell types, as only loosely mentioned in the text. Please clarify this point in methods and legends.

4) Presumably, motif enrichment was performed against a genomic background. This information is not found in the methods and should be added.

5) Motif enrichment analysis was done in windows of +200bp around DMRs. Please explain why this choice was made and whether a narrower window (e.g., 100bp) would yield drastically different outcomes.

6) Can the authors speculate on why they see enrichment for neuronal and pancreas gene ontology pathways related to conserved DNA methylation changes?

7) Data should be shown in important instances like "We identified 35 genes that were hypomethylated with aging and had close by an ARNT motif in all six cell types (Data not shown). Ten of these genes (right side of Figure 5C, genes under orange headings) have been linked to hypoxia response (37-46). Similarly, we found 20 genes with probes hypermethylated with age and with REST motif in the vicinity in all six cell types (data not shown)." Please provide the information.

---

## [Author Response]

Essential revisions:1) A number of points require clarification as indicated by the reviewers. Please address each of these points in revision.2) Clarify the study is a reanalysis of previously published data to shed light on new patterns.

We have addressed the points raised by the three reviewers. The data analyzed in this manuscript were collected in the context of the GESTALT study, a study of biomarkers of aging that is run in parallel to the Baltimore Longitudinal Study of Aging at the clinical site of the National Institute on Aging Intramural Research Program in Baltimore. The senior author of this manuscript designed the study and has been the π of this project from the beginning. Part of the GESTALT cohort data was previously deposited in the public domain as requested by the previous publication journal. In particular, the methylation data was previously used to explore epigenetic differences between different types of circulating immune cells in a previous published study [1]. However, methylation data was never used previously to look at association with aging, and to explore whether some CpG sites are differentially methylated with aging in the same direction across various circulating cell types. Identification of such shared CpG sites can suggest mechanisms of their agerelated changes. We note that it is common that data collected in a cohort study are used for multiple analyses that explore different hypotheses.

Reviewer #1 (Recommendations for the authors):This is an interesting study that potentially identifies a single fundamental mechanism that regulates chromatin changes in aging from stochastic changes. The notion that the methylation changes are cell selective is an interesting concept.Building on the comments regarding circulating and tissue-resident cells, can the authors provide data to support or refute whether programs are consistent in the different compartments? It would strengthen the work to provide discussion on these points.The interpretation of the data relies on changes in genes related to hypoxia to support that the metabolic state of the cell determines outcomes. It would be useful to test this hypothesis which would be possible in vitro to determine if programs are indeed determined by metabolic states that in vivo differ in different microenvironments, or that cells are intrinsically differentially programmed.Although the samples were obtained from participants who were deemed 'clinically' healthy, were any finer criteria used to distinguish and cross-correlate the immune cell phenotype with the methylation changes? The identification of an inflammatory program in the data associated with aging is interesting. While it might occur that low-level inflammation is inherently associated with aging, distinguishing the two programs would be important in teasing the data apart. Thus, a current limitation of the work is the lack of cellular phenotypic and functional mapping corresponding to the molecular analysis to bring clarity to whether overlapping programs occur and inflammation contributes to driving aging normally, or whether separate programs usually occur in parallel but could overlap. Dissecting this apart further would strengthen the work.

We thank the reviewer for the insightful suggestion. To investigate further into the role of age-associated inflammatory phenotype, we analyzed the published protein data from the same cohort of donors to look for age-associated changes in seven cytokines (IL6, IL1RN, IL1A, IL1B, TNF, TNFRSF1A, TNFRSF1B)[2]. For each cell type, all CpGs reported as significantly hypo- or hyper-methylated with age (in β-regression analyses adjusted for sex) were reanalyzed by separately adding each one of the seven pro-inflammatory cytokines Supplementary File 11. By comparing a model with a cytokine as explanatory variable (CpG ~ age + sex + cytokine) with another model without it (CpG ~ age + sex), we explored the robustness of age as explanatory variable of methylation, as well as possible mediating effects arising from pro-inflammatory cytokines.

In the Supplementary File 11, columns (C-J) show that the age association remains significant for most CpGs, even after adding one of these inflammatory mediators into the model. Interestingly we find that for age associated hypermethylated sites in CD4 cells, 10% of the CpGs associated with age show significant association with TNFRSF1A in the model which implies an association of these methylated sites or genes with the cytokine production and/or release in the circulation. The remaining columns (K-Q) show the number of CpGs for which there’s a significant association between CpG and the target analyte (in the model adjusted for age and sex). As noted above, TNFRSF1A and TNFRSF1B are associated with more than 7000 CD4 age-associated hypermethylated CpGs, suggesting an important role of age-differential methylation in the pro-inflammatory state of aging. For other cell types (most notably, B naïve and monocytes) TNF-α appears to play a role. Fewer associations are observed for the remaining analytes.

The new analysis has been added to the results and the description of the analysis has been added to the methods section.

Reviewer #2 (Recommendations for the authors):1. The authors claimed that clustering by cell type (PC2) was stronger than by age. For the PCA, it would be great if the authors could explain what PC1 indicates and present the information about all PCs.

We performed gene set enrichment analysis with the top 500 CpG probes corresponding to PC1, PC2 and PC3. Following this, the probes were annotated to nearby genes as per manufacturer’s annotation file and the genes which had probes with both positive and negative values for a PC were excluded. With the revised manuscript, we have provided a supplementary spreadsheet with columns providing information on the associated PC component and direction. A filter with q-value<1.e-5 was imposed to highlight the most significant pathways. PC1 genes are enriched for T cell receptor and Natural killer cells -associated pathways while PC2 shows enrichment of Hallmark IL2 STAT5 signaling and Reactome innate immune system. We conclude that PC1 contributes more towards separation of CD4, CD8 and NK cells from B cells, monocytes and granulocytes while PC2 separates the innate from the adaptive cells. The details have been added to the Results section (page 4) and the Methods section (page 18) along with the addition of Supplementary File 2.

2. It has been reported that sex-related differences contribute to immune cell aging (Huang et al., PNAS, 2021, PMID:34385315; Marquez et al., Nature Communications, 2020, PMID: 32029736). It is unclear how sex differences affect the dataset that the authors analyzed and how sex-adjusted β-regression was conducted.

We performed β regression with age and sex as two variables and in order to get the probes associated with age, we only selected the probes with adjusted p value for age <0.05 and excluded the ones that were also had p<0.05 for sex (N <1000 probes in total). We have added this information in greater detail in the Methods section (page 17). We also agree with the reviewers that identifying sex-differences is an interesting avenue for future research with greater number of samples.

3. The authors identified 350 age-associated differentially methylated sites among all six immune cell types and compared them with other published studies. Do these sites overlap with the DNA methylation clock sites identified in PBMCs since these sites change in the same direction in separate cell populations? Also, only 10 of these 350 sites overlap with Horvath's pan-tissue epigenetic clock. Is there any factor that can explain this difference? For example, sequencing depth, coverage, etc?

There can be several reasons for the observation that the CpGs identified in our study show only a small overlap with those identified in Horvath’s epigenetic clock. The “Horvath clock” was developed on methylation from multiple tissues and the “Hannum clock” was developed from whole blood. The contribution of different cell types depends on their frequency, and therefore methylation sites that change with aging in cells that are present in a high proportion would be predominant on those that only exist in low proportion. Indeed, the Horvath epigenetic clock works better in the GESTALT study when the data from PBMC are considered (59 probes overlapped instead of 10). Also, since these methylation clocks are statistically derived through a regression model using 21369 probes present in 27k microarray, it is possible that neighboring CpG gets picked up for the clock or many of the 850k sites went unrepresented. We have added this explanation to the Results section on page 5.

4. In the legend of Figure 2, the most significant genic probes are classified as -log10(padj)<10. Did the author mean -log10(padj)>10?

Thank you for pointing out the error. It has been fixed in the figure legend of the revised manuscript.

5. It needs to be clarified how the top 15 genes were chosen. How many age-associated CpGs contribute to these 15 genes? Also, if more genes are considered, will more genes be shared among cell types?

We thank the reviewer for the question. As a clarification we have incorporated additional information on number of age-associated probes for each of these top 15 genes in the Supplementary File 5. In Figure 1, we have looked at the overlap of all age-associated probes. Here we wanted to focus on whether the most significant age-associated genes are shared across cell types. When we increased the selection to the top 50 genes (new data added to Supplementary File 5), the number of shared age-hypo- or hypermethylated genes increase by only 5 and 13 genes respectively. This underscores our conclusion that “most CpGs with age-associated methylation consistent across cell types undergo moderate (although significant) methylation changes with aging.” We have rephrased this part of the Results on page 6 of the revised manuscript.

6. In Figure 3C and 3D, the differences in β coefficient in the InCHIANTI study are small (scale at 0.003 and 0.006). Are the differentially methylated sites called in GESTALT identified as differentially methylated in the original InCHIANTI study?

Changes in methylation with age (per year) tend to be small although they may be highly significant. Taking that into account, both the cross-sectional and longitudinal association with aging in the InCHIANTI are consistent in direction with those detected in the GESTALT study for the “shared” age-related CpG methylation.

7. In Figure 4C and Supp Figure 4, it seems both hypo- and hyper-methylated sites with age show the V-shaped pattern.

As correctly noted by the reviewer we do observe a V pattern and strong H3K4me1 peak at age-associated hypo as well as hypermethylated sites. This implies that the age-associated sites are enriched for active and/or primed enhancers. However, as the DHS and histone data in the ENCODE database was only available for either one of 2 donors (a 21-year-old male and a 37-year-old female), we were unable to see how the patterns change with age. We have added this note to the revised manuscript in the Results section (page 9) and Discussion section (page 13).

8. It is unclear how the expression levels were measured. Expression levels from qPCR or RNA-seq need to be shown in addition to Supp Table 6. RNA-seq was mentioned in the method section, and if this was performed in this study, the quality metrics for the data need to be provided and the data need to be uploaded to GEO. The current GEO number (GSE184269) provided in this manuscript is linked to a published study and does not contain RNA-seq data.

We thank the reviewers for their valid concern. We would like to clarify that expression values of the transcription factors are obtained from RNA-Seq data. In the paper we have specifically looked into the expression of 3 selected transcription factors to look for their association with age. Towards this, we have provided an additional Supplementary File 8 with the Kallisto TPM normalized values for the three transcription factors along with the sequencing data quality metrics.

9. There are several places where the authors mentioned "data not shown". It may be better if these data could be presented in supplementary figures or tables.

These are the 3 places where we have mentioned “data not shown” and all of them have been addressed-

1.“Both age-associated hypo- and hypermethylated sites showed evident H3K4me1 peaks, a marker commonly associated with active and primed enhancers (Figure 4C) (26). No specific trend was observed for H3K4me3 and H3K27ac (data not shown)”. We have added the results for H3K4me3 and H3K27ac to the Figure 4—figure supplement 1 and edited the Results section of revised manuscript (page 9).

2.“We identified 35 genes that were hypomethylated with aging and had close by an ARNT motif in all six cell types (Data not shown)”. We have added the results for all the 35 genes with ARNT motif in the vicinity, to the Supplementary File 10 and edited in the Results section of revised manuscript (page 11).

“Similarly, we found 20 genes with probes hypermethylated with age and with REST motif in the vicinity in all six cell types (data not shown)”. We have added the results for all the 26 genes with REST motif in the vicinity, to the Supplementary File 10 and edited in the Results section of revised manuscript (page 11). We also corrected a typing error as there are 26 genes and not 20 (page 11).

Reviewer #3 (Recommendations for the authors):Specific suggestions for Authors:1) As mentioned in the public review, certain claims should be toned down or placed in context.

In the revised version of the paper, we have added new citations and new data to address the concern raised by the reviewer.

2) The authors need to make it clear in the introduction that this current study is based on the reanalysis of a data set they have published in 2021 and describe key findings and limitations of the initial study.

We respectfully disagree with this statement. Very often data produced by cohort studies are reanalyzed according to different hypotheses. For example, NIA maintains the Baltimore Longitudinal study of aging and more than 1500 manuscript have been published over the last 20 years. Similarly, the genetic data from the UK biobank have been used for hundreds of analyses and produced some of the most interesting works in the literature over the last 10 years. There are probably hundreds of examples, including the Framingham study, the Cardiovascular Heart Study, the Health ABC study, and the Health and Retirement survey. In none of these cases, the content of previous papers is reported in the introduction, especially when, as in this case, the results are not relevant for the analysis reported in this manuscript.

3) It is not clear from the methods and figure legends that motif predictions were done on DMRs that could be found in at least 5 cell types, as only loosely mentioned in the text. Please clarify this point in methods and legends.

Thank you for your feedback. We used all the age-associated probes for HOMER analysis and not just the ones shared across 5 or more cell types. We added the associated details of the analysis on pages 17 and 20.

4) Presumably, motif enrichment was performed against a genomic background. This information is not found in the methods and should be added.

Thank you for pointing out the missing details. We added the HOMER background associated details in the Methods section on page 19. Briefly, HOMER software creates a list of background sequences which are randomly selected from the genome and matched for GC% content to the list of target sequences. We used this background file for our motif search in HOMER.

5) Motif enrichment analysis was done in windows of +200bp around DMRs. Please explain why this choice was made and whether a narrower window (e.g., 100bp) would yield drastically different outcomes.

We appreciate a very valid point raised by Reviewer 3. We have used a 200bp window because it is the default size in HOMER software. However following Reviewer 3 comments we reanalyzed the data with +50bp window size and the results have been tabulated in the revised Supplementary File 7. We observe that the top hits across 5 of the 6 cell types continue to be ARNT and CTCF for age-hypomethylated sites and REST for age-hypermethylated sites. Of note, unlike the hits with 200bp window size, Arid5a and Bcl6 were not observed in 50bp window size indicating that these motifs are comparatively farther from the CpG sites. We have added a note about the new analysis and the results on page 10 of the revised manuscript.

6) Can the authors speculate on why they see enrichment for neuronal and pancreas gene ontology pathways related to conserved DNA methylation changes?

We appreciate the question regarding the relevance of the abovementioned pathways to our study. A recent study by Ewing et al. found CpG sites getting hypermethylated in immune cells of multiple sclerosis patients mapped to neuronal pathways where they argue that neuronal genes like GRIN, GRID and Netrin play a critical role in immune cell development as well (https://pubmed.ncbi.nlm.nih.gov/31053557/). Also, a paper by Karagiannis et al. finds neuronal genes in their PBMC aging data. Both references have been added to the Results section (page 8). However, didn’t find much with respect to pancreatic genes in the literature warranting future investigation in this direction.

7) Data should be shown in important instances like "We identified 35 genes that were hypomethylated with aging and had close by an ARNT motif in all six cell types (Data not shown). Ten of these genes (right side of Figure 5C, genes under orange headings) have been linked to hypoxia response (37-46). Similarly, we found 20 genes with probes hypermethylated with age and with REST motif in the vicinity in all six cell types (data not shown)." Please provide the information.

Thank you for the suggestions. We have added the information in Supplementary File 10 along with modifying the Results section in the revised manuscript on page 11.